# Positive Association between the Use of Quinolones in Food Animals and the Prevalence of Fluoroquinolone Resistance in *E. coli* and *K. pneumoniae, A. baumannii and P. aeruginosa*: A Global Ecological Analysis

**DOI:** 10.3390/antibiotics10101193

**Published:** 2021-10-01

**Authors:** Chris Kenyon

**Affiliations:** 1HIV/STI Unit, Institute of Tropical Medicine, 2000 Antwerp, Belgium; ckenyon@itg.be; Tel.: +32-3-2480796; Fax: +32-3-2480831; 2Division of Infectious Diseases and HIV Medicine, University of Cape Town, Anzio Road, Cape Town 7700, South Africa

**Keywords:** *one-health*, food-animals, *E. coli*, *K. pneumoniae*, *Acinetobacter*, *P. aeruginosa*, fluoroquinolones, antimicrobial resistance, antibiotic consumption

## Abstract

(1) Background: It is unclear what underpins the large global variations in the prevalence of fluoroquinolone resistance in Gram-negative bacteria. We tested the hypothesis that different intensities in the use of quinolones for food-animals play a role. (2) Methods: We used Spearman’s correlation to assess if the country-level prevalence of fluoroquinolone resistance in human infections with *Acinetobacter baumannii*, *Escherichia coli*, *Klebsiella pneumoniae* and *Pseudomonas aeruginosa* was correlated with the use of quinolones for food producing animals. Linear regression was used to assess the relative contributions of country-level quinolone consumption for food-animals and humans on fluoroquinolone resistance in these 4 species. (3) Results: The prevalence of fluoroquinolone resistance in each species was positively associated with quinolone use for food-producing animals (*E. coli* [ρ = 0.55; *p* < 0.001], *K. pneumoniae* [ρ = 0.58; *p* < 0.001]; *A. baumanii* [ρ = 0.54; *p* = 0.004]; *P. aeruginosa* [ρ = 0.48; *p* = 0.008]). Linear regression revealed that both quinolone consumption in humans and food animals were independently associated with fluoroquinolone resistance in *E. coli* and *A. baumanii*. (4) Conclusions: Besides the prudent use of quinolones in humans, reducing quinolone use in food-producing animals may help retard the spread of fluoroquinolone resistance *in various Gram-negative bacterial species.*

## 1. Background

It is unclear why fluoroquinolone resistance in a range of bacterial species emerged so explosively in Asia over the past 20 years [1,2,3,4]. Between 1998 and 2009, for example, the prevalence of ciprofloxacin resistance in *Shigella* increased from 0% to 29% in Asia compared to 0% to 0.6% in Europe-America [1]. Likewise, fluoroquinolone resistance has emerged rapidly in other Gram-negative bacteria such as *Escherichia coli*, *Pseudomonas* spp., and *Klebsiella* spp. [1,2,3,4,5].

The emergence of fluoroquinolone resistance in various species of *Neisseria* is particularly instructive. The prevalence of gonococcal ciprofloxacin resistance in China increased from 10% in 1996 to 95% in 2003 [6]. By way of contrast the median prevalence of ciprofloxacin resistance in 2009 was 24% in the Americas and 6% in Africa [7]. In a similar vein, recent studies from China have found the prevalence of ciprofloxacin resistance to be 100% in commensal *Neisseria* and 66% in *N. meningitidis* [8,9,10].

There is however considerable heterogeneity in the prevalence of fluoroquinolone resistance within Asia and beyond. The prevalence of fluoroquinolone resistance in *N. gonorrhoeae* and *N. meningitidis* in Australia for example, is considerably lower than that in China [7,11,12]. Part of these differences may be explained by differences in fluoroquinolone consumption in humans [7]. Global ecological studies have however found that differences in fluoroquinolone consumption only explain a small proportion in the variation in fluoroquinolone resistance for organisms such as *E. coli* and *N. gonorrhoeae* [7,13,14]. This is also evident if we consider the examples of China and Australia where their not too dissimilar levels of consumption of fluoroquinolones appear to be an implausible explanation for the large differences in the prevalence in fluoroquinolones resistance in Gram-negative bacteria (Table 1). A more striking difference between China and Australia shown in Table 1, is the larger quantity of quinolones used for animal husbandry in China.

Quinolone use in food-producing animals has been linked to quinolone resistance in a number of Gram-negative pathogens circulating in humans [15,16]. This use of quinolones could induce resistance in bacteria circulating in humans both directly or indirectly. Direct selection would occur via human ingestion of quinolone residues in meat or water/soil contaminated by animal manure [17]. Quinolones have been found to show very low biodegradability in the environment [17,18]. Selection could also occur indirectly where quinolones select for resistance in bacteria in the food-animals and these bacteria or their resistance determinants are then transferred to humans. This indirect pathway has been shown to be important in the genesis and spread of cephalosporin resistance (mainly via the spread of plasmids) in various Gram-negative bacteria such as *E. coli* [15].

To the best of our knowledge, only one previous ecological analyses has assessed if there is an association between fluoroquinolone use in animal husbandry and fluoroquinolone resistance in human pathogens [19]. This study was limited to European countries and found that fluoroquinolone consumption in food-producing animals was positively associated with fluoroquinolone resistance in human infections with a number of Gram-negative pathogens including *Campylobacter jejuni* and *Salmonella* spp. [19,20]. Differences in fluoroquinolone use for food-producing animals are less pronounced between European countries than globally. We therefore hypothesized that differences in fluoroquinolone consumption in food-producing animals would be positively associated with a greater number of bacterial species than within Europe.

## 2. Methods

### 2.1. Data

#### 2.1.1. Antimicrobial Resistance Data

The country-level prevalence of fluoroquinolone resistance for *A. baumannii, Escherichia coli, Klebsiella pneumoniae* and *Pseudomonas aeruginosa* was taken from the Center for Disease Dynamics, Economics & Policy’s (CDDEP) ResistanceMap database. CDDEP aggregates data on antibiotic resistance from several sources. The isolates tested are all invasive taken from blood/cerebrospinal fluid from humans. The data are then harmonized to present similar definitions of resistance across countries and regions to enable comparisons between countries. Further details pertaining to the methodology and definitions used to define antimicrobial resistance can be found at [21]. The list of sources used to obtain the data is provided in Appendix A. CDDEP provides data on fluroquinolone resistance for 10 bacterial species but only 4 of these have data for more than 15 countries. We limited our analyses to these 4 species. For each of the species, a resistance prevalence estimate from a single year for each country was provided in the dataset. This typically applied to the year 2017.

#### 2.1.2. Quinolone Use for Food-Animal Data

We obtained the country level consumption of quinolones for animal food production in the year 2013 from a systematic review on this topic performed by Broeckel et al. [22]. This study calculated the volume of antimicrobials (in tons) by class of antimicrobial in 38 countries in the year 2013. Four categories of animals were included: chicken, cattle, pigs and small ruminants (sheep and goats), which together account for the overwhelming majority of terrestrial animals raised for food [15,22]. 

We used this data to calculate the number of milligrams of quinolones used for animal food production/population correction unit (PCU—a kilogram of animal product) in the year 2013. The data for the tonnage of food animals produced per country and year in the year 2013 was taken from the Food and Agriculture Organization estimates (http://www.fao.org/faostat/en/?#data/ accessed on 2 July 2021).

#### 2.1.3. Human Fluoroquinolone Consumption Data

Data from IQVIA (Quintiles and IMS Health) were used as a measure of national antimicrobial drug consumption in 2015—the most recent year for which data is available. Details for how IQVIA calculates these consumption estimates is provided in Appendix A [7].

### 2.2. Statistical Analyses

For each comparison, Spearman’s correlation was used to assess the country-level association between the prevalence of fluoroquinolone resistance in each species and (1) quinolone use for animals and (2) quinolone consumption by humans. Linear regression was used to assess the country-level association between the prevalence of fluoroquinolone resistance in each species and the two independent variables in 3 models. In the first model, we assessed the association between fluoroquinolone resistance and fluoroquinolone consumption in humans (Model-1). In Model-2 we assessed the association between fluoroquinolone resistance and quinolone use in animals. In Model-3 we evaluated the effect of both independent variables on fluoroquinolone resistance.

### 2.3. Sensitivity Analysis

China had a considerably higher consumption of quinolones for food-producing animals which meant it was a clear outlier in the dataset and may have skewed the linear regression analyses (Table 1 and Appendix A). In sensitivity analyses, we therefore repeated the analyses excluding China. All statistical analyses were performed in Stata 16.0 and a *p*-value of <0.05 was regarded as statistically significant.

## 3. Results

The prevalence of fluoroquinolone resistance varied considerably between countries (*E. coli*- median 31.5% [interquartile range (IQR) 22.5–47.5]; *K. pneumoniae*- median 44% [IQR 27–62]; *A. baumannii*- median 53.5% [IQR 14–82]; *P. aeruginosa*- median 21.5% [IQR 15–34]; Table 1). Large differences in the consumption of fluoroquinolones were also evident between countries-median 721 defined daily doses/1000 inhabitants/year (IQR 421–1129; Table 1).

Quinolone use for food-producing animals varied considerably in the 36 countries with data available (median 1.9 mg quinolones/PCU (IQR 0.7–6.6 mg/PCU; Table 1). Quinolone exposure was highest in China (261.2 mg/PCU).

### 3.1. Spearman’s Correlations

The prevalence of fluoroquinolone resistance in each species was positively associated with quinolone use for food-producing animals (*E. coli* [ρ = 0.55; *p* < 0.001; *n* = 35], *K. pneumoniae* [ρ = 0.58; *p* < 0.001; *n* = 31]; *A. baumannii* [ρ = 0.54; *p* = 0.004; *n* = 26]; *P. aeruginosa* [ρ = 0.48; *p* = 0.008; *n* = 29]) and quinolone consumption in humans (*E. coli* [ρ = 0.58; *p* < 0.001; *n* = 47], *K. pneumoniae* [ρ = 0.42; *p* = 0.006; *n* = 42]; *A. baumannii* [ρ = 0.54; *p* < 0.001; *n* = 54]; *P. aeruginosa* [ρ = 0.58; *p* < 0.001; *n* = 37]; Table 2).

### 3.2. Linear Regression Models

For both *K. pneumoniae* and *P. aeuginosa*, only human consumption of fluoroquinolones had a statistically significant effect on the prevalence of resistance (Table 3). In the case of *E. coli* and *A. baumannii*, both consumption in humans and food animals were significantly associated with fluoroquinolone resistance (Table 2). In the case of *A. baumannii*, this association was statistically significant in the multivariate but not the bivariate model. For both species, the combined model (Model-3) was a better predictor of fluoroquinolone resistance than Model-2 which only considered human fluoroquinolone consumption (*E. coli*: R^2^ increased from 0.27 to 0.48; *A. baumannii*: R^2^ increased from 0.26 to 0.59; Table 2).

### 3.3. Sensitivity Analyses

Excluding China from the Spearman’s correlations had no effect on the results (Appendix A). It did however affect the results of the linear regression analyses. The major change was that the positive association between the prevalence of fluoroquinolone resistance in *E. coli* and the consumption of quinolones in food-producing animals was no longer statistically significant (Appendix A).

## 4. Discussion

In this global ecological study Spearman’s correlation revealed that the prevalence of fluoroquinolone resistance in all four species was positively associated with the use of quinolones for food-animals. In the case of *E. coli* and *A. baumannii,* linear regression analyses suggested that quinolone consumption in both humans and food animals plays a role in the explaining global differences in the prevalence of fluoroquinolone resistance. As far as *K. pneumoniae* and *P. aeruginosa* were concerned, this association was statistically significant in the Spearman’s correlation but not the linear regression analyses. This difference is likely influenced by one outlier in the data—China. In the dataset, China has a very high consumption of quinolones for food animals, a high prevalence of resistance for *E. coli, A. baumannii* and lower resistance prevalences for *P. aeruginosa* and *K. pneumoniae*. The results of the sensitivity analysis are compatible with this explanation.

Numerous limitations mean that due caution should be exercised in drawing conclusion from this analysis. These limitations include the relatively small number of countries with available data, the lack of longitudinal data on quinolone consumption in animals and the absence of data on quinolone use for aquaculture. National differences in the minimum time between last quinolone administration and slaughter may also influence the relationship between quinolone consumption and induction of quinolone resistance. The fluoroquinolone resistance prevalence estimates from CDDEP are based on various methodologies making cross country comparisons problematic. We did not adjust our analyses for either differences in susceptiblity testing strategies or breakpoints between countries or over time as this information is not provided by CDDEP. These limitations should however result in a misclassification bias which would typically result in a bias towards the null hypothesis [23]. The epidemiology of resistance is complex and factors other than the amount of quinolones consumed may influence the level of quinolone resistance. These include poor sanitation, inadequate processing of sewage, substandard regulation of antimicrobials, weak antimicrobial stewardship, consumption of other classes of antimicrobials, travel by humans and trade of live animals and meat, variations in environmental temperatures and high levels of institutional corruption [3,4,5,13,14,15,24]. We did not control for any of these.

Despite these limitations, various types of evidence suggest that excessive use of antimicrobials in food-producing animals could play a role in inducing antimicrobial resistance in bacteria in humans. In addition to the ecological evidence of a positive association between quinolone consumption for food animals and fluoroquinolone resistance in bacteria in humans from Europeans countries reviewed above [19], other European studies have found positive assocations between the prevalence of fluoroquinolone resistance in *E. coli* in humans and *E. coli* from poultry and pigs [20]. A systematic review on the topic found evidence that fluoroquinolone and cephalosporin resistance could be transferred from *E. coli* in food-producing animals to humans [16].

As noted above, an alternative pathway for quinolones used in food-animal production to induce resistance would be via humans ingesting quinolone residues in meat or water/soil contaminated by animal manure [17]. Antimicrobial concentrations up to 230-fold lower than the minimal inhibitory concentration can induce antimicrobial resistance in bacteria such as *E. coli* and *Salmonella enterica* spp. [25,26]. Concentrations of ciprofloxacin as low as 0.1 μg/L have, for example, been shown to be able to select for resistance in certain Gram-negative bacteria [25,27]. This is termed the minimum selection concentration (MSC) [25]. Quinolone concentrations in meat, water and environmental samples exceed this threshold by some margin in a number of countries, but especially so in certain Asian countries. For example, studies have found that the mean concentration of ciprofloxacin in samples of milk, eggs, and edible fish in China to be 8.5 µg/L, 16.8 µg/kg and 331.7 µg/kg, respectively, [28,29,30]. The ingestion of these relatively high concentrations of quinolones in food products was the favoured explanation for the the high median concentration of quinolones (median 20 μg/kg), found in the colons of the general human population in 3 regions of China [31]. This concentration is 200 fold higher than the MSC for *E. coli* [27]. In a similar vein, a study from South Korea found that high urinary excretion of enrofloxacin and ciprofloxacin in the general population were strongly associated with consumption of beef, chicken and dairy products [32]. Finally, reducing the consumption of these foodstuffs in South Korea has been shown to result in a reduction of urinary quinolone concentrations [33]. Very low concentrations of antimicrobials such as fluoroquinolones can not only generate de novo resistance but they can also select for the enrichment of already present resistant mutants [25,27]. It is thus possible that fluoroquinolone consumption in humans plays a dominant role in the genesis of de novo resistance and that low concentrations of quinolones consumed in food may promote the spread of these resistant strains.

Ecological studies are best considered hypothesis generating. The results of this study need to be followed up by detailed individual-level, association studies. Randomized controlled trials would be particularly valuable. One study design would be to randomize groups of mice or humans to various schemas of low dose antimicrobials to assess the lowest dose of an antimicrobial that does not select for resistance in resident bacteria.

## Figures and Tables

**Table 1 antibiotics-10-01193-t001:** Country-level consumption of quinolones for food-producing animals (milligrams of quinolones used for animal food production/PCU), fluoroquinolone consumption in humans (defined daily doses[DDD]/1000 inhabitants per year) and prevalence of resistance to fluoroquinolones (%) for 4 bacterial species.

Country	Quinolone Food Animals (mg/PCU)	Quinolones Humans (DDD/1000 inh./yr)	*K. pneumoniae* (%)	*E. coli* (%)	*P. aeruginosa* (%)	*A. baumannii* (%)
Argentina		648	51	34	25	83
Australia	0	245	4	12	5	6
Austria	0.0706	699	16	22	12	9
Belarus			83	48	87	90
Belgium	0.2882	1246	27	25	12	14
Bosnia			56		40	
Bulgaria	0.3877	1178	68	43	32	96
Canada	0.0105	815	19	21	19	
Chile		577		29		
China	16.0223	389	42	56	15	82
Croatia		733	48	29	38	98
Cyprus	0.1139		45	44	25	76
Czech Republic	0.1149	402	54	26	30	20
Denmark	0.3004	292		14		1
Ecuador		733	47	59	22	55
Egypt		1152	80			
Estonia	0.1554	376	30	20	13	
Finland	0.0184	398	15	14	11	3
France	0.1415	688	29	17	17	13
Georgia			68			
Germany	0.1414	701	18	23	16	9
Ghana			61	59	29	26
Greece		1242	69	34	38	96
Hungary	0.7233	1130	42	31	23	67
Iceland	0.0121			14		
India		762	69	84	34	58
Ireland	0.1945	433	19	26	16	
Italy	0.5165	1486	58	47	29	79
Japan	0.0518	954		30		
Kenya			40	58		
Latvia	0.1951	418	35	32		
Lebanon				45		
Lithuania	0.1657	564	66	28	21	
Luxembourg	0.1508	1064	29	.	21	
Malawi			45	.		
Malaysia		355		26		50
Mexico		434	28	62	23	95
Netherlands	0.1671	374	16	16	12	3
New Zealand	0.0088	136		10		
Nigeria			75	76		
Norway	0.1408	258	13	16	5	0
Oman			42	43		
Pakistan		1642	58	59		
Philippines		262	32	39	18	40
Poland	0.887	638	68	38	39	83
Portugal	0.7636	876	49	30	25	38
Romania		1382	66	28	62	90
Russia		1129	87	63	58	94
Saudi Arabia			40	47		
Serbia		1158	74		53	
Slovakia	0.1339	1088	68	47	47	52
Slovenia	0.1462	546	34	26	20	50
South Africa		589		28	35	66
South Korea	1.0048	766		37		
Spain	1.464	1180	24	33	24	73
Sri Lanka	0.1531	578	49	59		33
Sweden	0.00418	354	12	17	9	0
Switzerland	0.05	716	11	19	7	14
Tajikistan				.	17	
Thailand		726	35	47	15	57
Tunisia		851	57	19		
Turkey		1352	62	55	35	92
UAE		1280	27	49		
USA	0.0538	1002	10	31	19	39
United Kingdom	0.038	251	12	18	10	17
Venezuela		1199	17	50		81
Vietnam	0.1196	1162	44	66	21	57
Zambia			69			
Zimbabwe			44			

**Table 2 antibiotics-10-01193-t002:** Spearman’s correlation matrix of country-level prevalence of fluoroquinolone (FQ) resistance (%) in 4 bacterial species and quinolone consumption in food-producing animals (milligrams of quinolones used for animal food production/PCU) and fluoroquinolone consumption in humans (defined daily doses/1000 inhabitants per year).

	*Acinetobacter baumannii*	*Escherichia coli*	*Pseudomonas aeruginosa*	*Klebsiella pneumoniae*	Food-Animal FQ Consumption	Human FQ Consumption
** *Acinetobacter baumannii* **	1					
** *Escherichia coli* **	0.66 **	1				
** *Pseudomonas aeruginosa* **	0.76 **	0.72 **	1			
** *Klebsiella pneumoniae* **	0.72 **	0.61 **	0.90 **	1		
** *Food-animal FQ consumption* **	0.54 **	0.55 **	0.46 **	0.58 **	1	
**Human FQ consumption**	0.54 **	0.58 **	0.58 **	0.42 *	0.35 *	1

* *p* < 0.05 ** *p* < 0.005; FQ—fluoroquinolone.

**Table 3 antibiotics-10-01193-t003:** Linear regression models testing the country-level association between quinolone consumption in food-producing animals and humans and the prevalence fluoroquinolone resistance in *E. coli, K. pneumoniae, A. baumannii* and *P. aeruginosa* spp. [coefficients (95% confidence intervals)].

	*E. coli*			*K. pneumoniae*			*A. baumannii*			*P. aeruginosa*		
	Model 1	Model 2	Model 3	Model 1	Model 2	Model 3	Model 1	Model 2	Model 3	Model 1	Model 2	Model 3
Quinolones food animals	1.93 (0.21–3.65) *	-	2.2 (0.84–3.56) **	0.84 (−1.76–3.44)	-	1.21 (−1.25–3.68)	3.62 (−0.39–7.64)	-	4.6 (1.79–7.46) **	−0.11 (−1.42–1.21)	-	0.14 (−1.02–1.30)
Quinolones humans	-	0.02 (0.01–0.03) **	0.02 (0.01–0.03) **	-	0.02 (0.01–0.04) **	0.02 (0.00–0.04) *	-	0.05 (0.02–0.08) **	0.06 (0.03–0.08) **	-	0.02 (0.01–0.03) **	0.01 (0.01–0.02) **
*n*	35	47	33	31	42	30	26	35	25	29	37	28
R^2^	0.14	0.27	0.48	0.01	0.19	0.18	0.13	0.26	0.59	0.00	0.29	0.29

* *p*-value < 0.05, ** *p*-value < 0.005.

## Data Availability

The data we used is publicly available from: https://resistancemap.cddep.org/AntibioticResistance.php (accessed on 2 July 2021).

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
