# Peer review of "Positive Association between the Use of Quinolones in Food Animals and the Prevalence of Fluoroquinolone Resistance in E. coli and K. pneumoniae, A. baumannii and P. aeruginosa: A Global Ecological Analysis"

_antibiotics, 2021, doi:10.3390/antibiotics10101193_

Round 1
Reviewer 1 Report
The authors investigated the correlation between the consumption of quinolones by humans and livestock and antibiotic resistance in selected gram negative bacteria worldwide. The study is compact, well-written, statistically sound, and the results are presented and discussed objectively. I have only a view comments that I wrote directly to the PDF file (see attachment).

Author Response
Reply to Reviewer 1:
We thank the reviewer for their most useful comments and suggestions.
Not only in food-producing animals but also in humans. Please do not exclude this aspect in the abstract, if only in a subordinate clause (for example, "Besides the prudent use of quinolones in human therapy, reducing...").
Reply:
The abstract has been edited according to the most useful suggestion of the reviewer.
Throughout the text: A. baumannii.
Reply:
The spelling of A. baumannii has been corrected throughout the manuscript
It plays an important role where the isolates originate from (humans/animals//clinical/healthy). Has this aspect been taken into account? Are the data comparable considering this fact? It is laborious to research the data from the individual records - please provide information (at least on isolates from China and Australia in the discussion section due to the great differences).
Reply:
The methods section has been expanded to make the origin of the isolates clearer (P 4, L4 onwards):
CDDEP aggregates data on antibiotic resistance from several sources. The isolates tested are all invasive taken from blood/cerebrospinal fluid from humans. The data are then harmonized to present similar definitions of resistance across countries and regions to enable comparisons between countries. Further details pertaining to the methodology and definitions used to define antimicrobial resistance can be found at [21].
Please provide information on the units not only in the table heading, but also in the table itself.
Reply:
This has been done.
How can this result be explained? Please discuss.
Reply:
There are a number of possible explanations for this finding. i) Nosocomial antimicrobial exposure plays a bigger role in the genesis of antimicrobial resistance for A. baumannii and P. aeruginosa than E. coli and K. pneumoniae. This involves the fact that these infections are more likely to be nosocomially acquired and transmitted. ii) Whilst the effects of quinolone consumption for food animals were not statistically significant predictors of quinolone resistance in K. pneumoniae and P. aeruginosa, the coefficients were positive in both cases. It may therefore be that a bigger sample size or samples with less misclassification bias may have revealed a statistically significant effect. iii) The significant associations we found between quinolone consumption in food animals and resistance in E. coli and A. baumannii may be false positives.
Larger studies with standardised methodologies for sampling and assessment of antimicrobial susceptibility would be useful to ascertain if one these explanations is correct.
Be
Reply:
Thank you for pointing out this typo which has been corrected.
What about the withdrawal time? Is it different in different countries (or even non-existent? Quinolone residues might strongly depend on this period.
Reply:
Thank you for pointing this out. We have added the following text to the limitations section to address this issue (P7, Discussion section, second paragraph):
National differences in the minimum time between last quinolone administration and slaughter may also influence the relationship between quinolone consumption and induction of quinolone resistance.
However, this study would also be limited since we cannot estimate the extent of transmission in reality (the fact that the use of antibiotics will lead to an increased resistance prevalence is well-known).
Reply:
Our wording here was suboptimal. We have changed the wording to the following to make this point clearer (P 8, last sentence):
One study design would be to randomize groups of mice or humans to various schemas of low dose antimicrobials to assess the lowest dose of an antimicrobial that does not select for resistance in resident bacteria.
Reviewer 2 Report
Overall, a sound study. It is well known that antibiotic consumption in humans and agricultural use varies by country. This study examines the hypothesis that antibiotic consumption in humans and food animals is associated and correlates with the prevalence of antibiotics. The author uses data from existing and available databases to test the hypothesis, and looks to find these associations with straightforward statistical analysis. The author submits a well-written manuscript reporting the findings, which are not surprising, but nonetheless significant. The author acknowledges the hypothesis-generating nature of the results, as well as other limitations. I appreciated the discussion section: "The epidemiology of resistance is complex and factors other than the amount of quinolones consumed may influence the level of quinolone resistance...."
Minor edits:
- define all abbreviations and acronyms at first mention (for example, AMR, IQVIA, etc)
- "Pearson's" correlation should be "Spearman's" in the "Sensitivity Analyses" in results section
Author Response
Reply to Reviewer 2:
We thank the reviewer for their most useful comments and suggestions.
Minor edits:
- define all abbreviations and acronyms at first mention (for example, AMR, IQVIA, etc)
Reply:
This has been done.
- "Pearson's" correlation should be "Spearman's" in the "Sensitivity Analyses" in results section
Reply:
This change has been made.